# Piezoceramic membrane with built-in ultrasound for reactive oxygen species generation and synergistic vibration anti-fouling

Yang Zhao [1,2] ✉, Feng Yang[1,2], Han Jiang[1,2] & Guandao Gao [3,4]

Piezoceramic membranes have emerged as a prominent solution for membrane fouling control. However, the prevalent use of toxic lead and limitations of vibration-based anti-fouling mechanism impede their wider adoption in water treatment. This study introduces a $Mn/BaTiO_3$ piezoceramic membrane, demonstrating a promising in-situ anti-fouling efficacy and mechanism insights. When applied to an Alternating Current at a resonant frequency of 20 V, 265 kHz, the membrane achieves optimal vibration, effectively mitigating various foulants such as high-concentration oil (2500 ppm, including real industrial oil wastewater), bacteria and different charged inorganic colloidal particles, showing advantages over other reported piezoceramic membranes. Importantly, our findings suggest that the built-in ultrasonic vibration of piezoceramic membranes can generate reactive oxygen species. This offers profound insights into the distinct anti-fouling processes for organic and inorganic wastewater, supplementing and unifying the traditional singular vibrational anti-fouling mechanism of piezoceramic membranes, and potentially propelling the development of piezoelectric catalytic membranes.

The escalating global need for efficient wastewater treatment and access to purified drinking water has driven significant advancements in water treatment technologies[1,2]. Membrane-based separation, renowned for its energy-saving and efficacy, plays a pivotal role in meeting these requirements[3,4]. However, membrane fouling, characterized by organic and inorganic micro-nano particles commonly recognized as the Achilles' heel of membrane separation technology, poses a major challenge by adversely affecting membrane flux, energy efficiency, and membrane lifespan[3–5]. Consequently, considerable efforts have been directed towards devising effective anti-fouling strategies, such as pre-treatment of raw water, operational condition optimization, chemical cleaning, and modifying the membrane surface with static charges to eliminate or repel foulants from adhering to the membrane[6,7]. Regrettably, these traditional anti-fouling methods often necessitate extra processing steps or brief operational pauses and are generally limited to removing foulants, which can also degrade the structural integrity of membranes[6,8]. To overcome the limitations of traditional passive anti-fouling methods, active or dynamic anti-fouling strategies have emerged as a viable approach[3,6,9]. Active anti-fouling strategies, especially those utilizing continuous ultrasonic vibration or photocatalytic activities, have demonstrated notable effectiveness[10–12]. However, issues like membrane deterioration, the necessity for cavitation regulation, and high energy consumption still limit their practical applications[8,11]. Fortunately, these drawbacks of

[1]School of Energy and Environment, Southeast University, Nanjing 210096, China. [2]State Key Laboratory of Environmental Medicine Engineering, Ministry of Education, Southeast University, Nanjing 210096, China. [3]State Key Laboratory of Pollution Control and Resource Reuse, School of Environment, Nanjing University, Nanjing 210023, China. [4]Research Center for Environmental Nanotechnology (ReCENT), Nanjing University, Nanjing 210023, China. ✉e-mail: zhaoyangseu@seu.edu.cn

conventional ultrasonic approaches can be overcome through the use of self-vibrating piezoelectric membranes that harness the inverse piezoelectric effect[13,14]. The inverse piezoelectric effect, which converts Alternating Current (AC) stimulation into built-in membrane ultrasonic vibrations, has been conclusively shown to mitigate fouling, thus improving membrane flux and extending its operational life[13,14].

Organic piezoelectric response membranes, notably polyvinylidene fluoride (PVDF) types, have been recognized for their hydrophilicity, cost-effectiveness, and efficient anti-fouling via built-in membrane ultrasonic vibration[15]. Their disadvantage lies in the typically low piezoelectric coefficient ($d_{33} \leq 30$ pC N$^{-1}$), limited durability to stubborn foulants like oil and bacteria[16,17]. These limitations highlight the need for alternative piezo-materials that offer enhanced antifouling performances with durability[14,18]. Ceramic membranes offer superior chemical stability and mechanical strength. Typically, the piezoceramics could self-clean by in-situ vibration under AC stimulation[14,17]. However, the environmental and health risks about lead leakage from commonly used lead zirconate titanate (PZT) limit the use of piezoceramic membranes in anti-fouling[19,20]. Recent progress in lead-free piezoceramics, including quartz-based materials, presents possibilities for fouling control. However, these are limited by their lower piezoelectric coefficients (commonly $\leq 10$ pC N$^{-1}$) and the requirement for high driving voltages[21]. Additionally, the basic physical vibration anti-fouling mechanism in piezoelectric membranes may not fully address organic fouling, especially for adhesive organic contaminants, typically necessitating chemical approaches like oxidation[3,22]. Therefore, there is an urgent need to develop lead-free piezoceramic membranes that deliver effective piezoelectric responses for comprehensive anti-fouling applications and to conduct thorough investigations into the mechanisms of anti-fouling[18,23]. Barium titanate (BaTiO$_3$) emerges as a leading candidate in lead-free piezoceramics, exhibiting superior piezoelectric properties ($d_{33} \geq 100$ pC N$^{-1}$)[3,16]. Enhanced further by strategic elemental doping, its piezoelectric capabilities and mechanical robustness can be significantly improved, paving the way for more effective anti-fouling applications[3,24]. Importantly, BaTiO$_3$ is safe and has been reported for use in personal care and medical fields such as dental cleaning and cell tissue culture[25–27]. In prior research, we harnessed the inherent hydraulic pressure of hydraulically driven membrane processes to elicit the direct piezoelectric effect in the Mn-doped BaTiO$_3$ (Mn/BaTiO$_3$) piezoceramic membrane, thereby controlling fouling[3]. This piezoceramic membrane transforms periodic pressure fluctuations into electrical pulses and rapid voltage changes, thus generating reactive oxygen species (ROS) and dielectrophoretic (DEP) forces at the membrane surface. ROS degrades foulants via oxidation or severs their attachment to the membrane surface, followed by DEP force driving the foulants away from the membrane surface. Since DEP force is independent of the type of foulants, this mechanism offers a universal approach to fouling control[3]. Notably, the inverse piezoelectric effect, corresponding to the direct piezoelectric effect, is also crucial in anti-fouling applications via piezoelectric ultrasonic vibration. Since at least 1992, there have been reports of using piezoelectric ultrasonic vibration for membrane fouling mitigation. Despite decades of research, this remains the sole mechanism identified, lacking further breakthroughs[23,28,29]. Actually, from an ultrasonic perspective, the core component of an ultrasonic generator is the piezoelectric ceramic or piezoelectric crystal, and the built-in ultrasonic resonant frequency of piezoceramic membranes usually falls in the high-frequency ultrasonic range (>100 kHz). This high-frequency can generate significant ROS for advanced oxidation processes in water treatment, which previous studies overlooked but is potentially crucial for fouling control in piezoceramic membranes[3,14,30,31].

In this study, through the Mn/BaTiO$_3$ piezoceramic membrane, we comprehensively discussed the anti-fouling performances and potential mechanism insights of piezoceramic membranes via the inverse piezoelectric effect, as illustrated in Fig. 1a, b. An in-depth analysis was conducted to determine the optimal operating conditions for the membrane. Additionally, the study investigated the essential antifouling mechanisms of membranes, focusing on built-in piezoelectric ultrasonic vibration and ROS synergy, and combined theoretical simulations with experiments across diverse wastewater scenarios.

## Results

### Membrane characterization
The Mn/BaTiO$_3$ piezoceramic membrane, depicted in Fig. 1c (insert), was fabricated using a sintering method (Fig. 1a)[3,32]. This membrane is characterized by its porous surface and detailed cross-sectional microstructure, as shown in the Environmental Scanning Electron Microscopy (ESEM) images, Fig. 1c–e, g–i and Supplementary Fig. 1a. It possesses an average pore size of 210 nm (Supplementary Fig. 1b), a bulk porosity of 22.43%, and measures 30 mm in diameter with a thickness ranging from 2 to 2.2 mm. This membrane exhibits hydrophilic properties, as indicated by an underwater oil angle exceeding 140° (Supplementary Fig. 1c, d) and a pure water permeability of ~91 LMH bar$^1$, which correlates with membrane thickness.

High-Angle Annular Dark-Field (HAADF) imaging and Energy-Dispersive X-ray Spectroscopy (EDS) mapping of the sintered grains (Fig. 1f) and membrane surface (Supplementary Fig. 1a) confirm the composition of Ba, Ti, O, and Mn, derived from the base material of BaTiO$_3$ and the Mn$_2$O$_3$ sintering additive. In addition, as evidenced by the XRD patterns (Supplementary Fig. 1e), the addition of trace amounts of Mn$_2$O$_3$ resulted in only a negligible alteration of the host BaTiO$_3$ crystal structures, with the exception of an enhanced split between the peaks of (002) and (200) at 2θ around 45°. This distinct split signifies the presence of the piezoelectric tetragonal phase in the solid-state sintered BaTiO$_3$ (Fig. 1c, insert), a distinctive feature of the piezoelectric properties of barium titanate[32,33].

### Self-cleaning performance
We assessed the optimal self-cleaning conditions for the piezoceramic Mn/BaTiO$_3$ membrane in a cross-flow filtration system, applying different AC settings (as depicted in Fig. 2a and Supplementary Fig. 2), and using a typical oily wastewater O/W (Oil-in-Water, <d> = 0.97 μm, see Fig. 3a, insert) emulsion as the feed solution[34]. The AC frequency and voltage significantly influenced the normalized membrane flux, as illustrated in Fig. 2b, c. The Mn/BaTiO$_3$ piezoelectric membrane, when operated at a frequency of 265 kHz, maintained a remarkably stable water permeance, retaining 85.9 - 91.0% of its initial value during oil emulsion treatment. In stark contrast, membranes vibrated with AC frequencies of 100, 600, and 5 kHz, maintaining only 41.4 - 80.4% efficiency after 180 min of operation. It was observed that within a certain range of the tested AC frequencies, membrane anti-fouling effectiveness did not proportionally increase with higher AC frequencies[14]. This suggests that the AC frequency of 265 kHz is optimal for generating substantial piezoelectric vibration in the Mn/BaTiO$_3$ membrane, leading to consistently high oil anti-fouling efficiency. Furthermore, increasing the intensity of AC, particularly at a resonance frequency of 265 kHz, amplified the vibrational amplitude of membrane. As a result, when the AC voltage was raised from 0 to 20 V, there was a corresponding enhancement in anti-fouling performance, ranging from 38.3% to 91.0% (Fig. 2b, c).

Post-filtration analysis using ESEM and elemental carbon mapping revealed significant oily deposits on the surface and internal cross-section of the Mn/BaTiO$_3$ piezoelectric membrane when no AC voltage was applied (0 V), in stark contrast to the condition with 20 V AC voltage (Fig. 2d, e, Supplementary Fig. 3). Regarding membrane fouling, reversible fouling, usually resulting from sparse oil deposits on the membrane surface, can often be mitigated with methods such as flushing to restore flux. Conversely, fouling that penetrates the membrane surface and its pores, causing irreversible contamination,

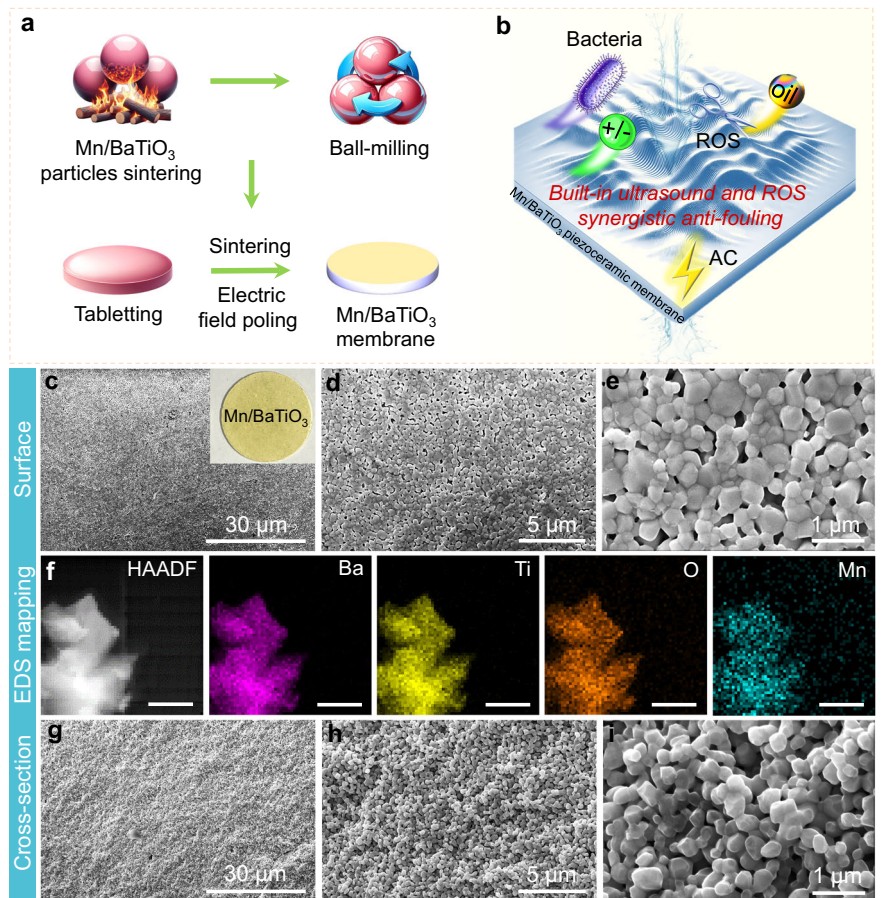

**Fig. 1 | Fabrication, characterization and anti-fouling mechanism insights of the Mn/BaTiO₃ piezoelectric membrane. a** Membrane fabrication process. **b** Anti-fouling mechanism insights. **c–e** ESEM images of the surface (with an optical image insert in (**c**) and **g–i** cross-section of the membrane. **f** STEM-HAADF image with EDS elemental mapping of Mn/BaTiO₃ grains, scale bar, 100 nm.

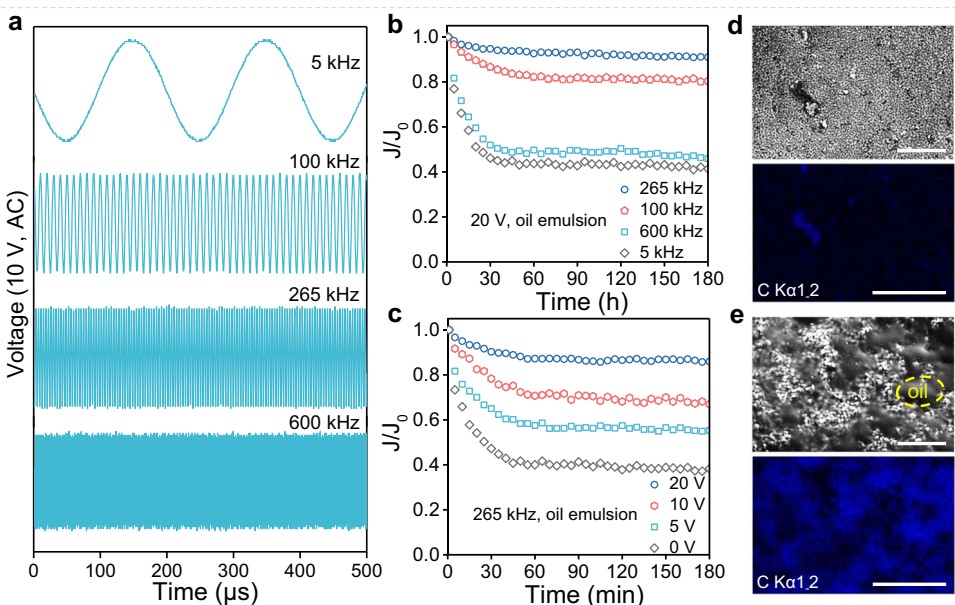

**Fig. 2 | Oil fouling control of the Mn/BaTiO₃ piezoelectric membranes. a** AC frequency variation (5–600 kHz, example at 10 V). **b, c** Time-based normalized membrane flux under different AC conditions for oil emulsion treatment. Comparative ESEM and the corresponding EDS elemental mapping images of oil-fouled Mn/BaTiO₃ piezoelectric membranes surface at 20 V, 265 kHz (**d**) and 0 V (**e**), scale bar: 5 μm.

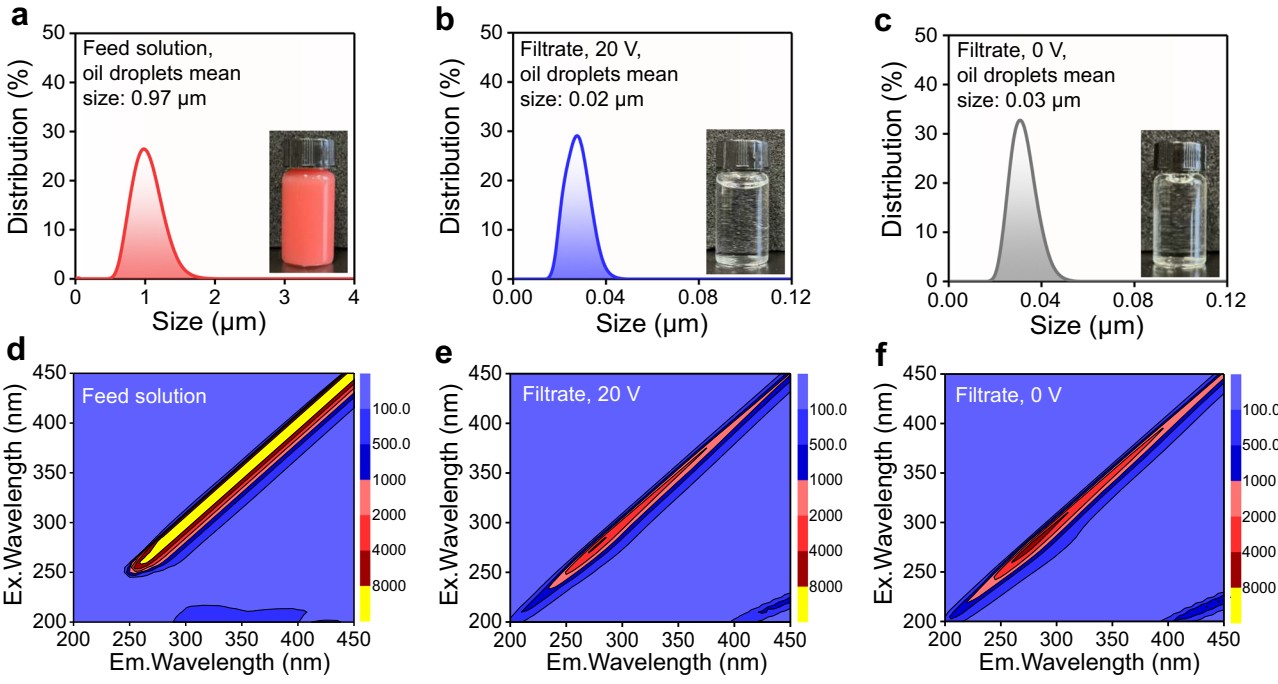

**Fig. 3 | Membrane retention results.** Particle size distributions of the colored oil emulsion before (**a**, 2500 ppm) and after (**b**, **c**) Mn/BaTiO$_3$ piezoelectric membrane treatment (optical images of water samples, inset). FEEMs analysis of feed (**d**) and filtrates (**e**, **f**, filtration duration of 180 min) across different filtration processes.

significantly compromises membrane performance[8,35]. Although AC stimulation did not completely eliminate membrane fouling over the within the 180 min test period, it significantly enhanced the flux recovery ratios (*FRR*), achieving 99.1% at 20 V (reversible fouling ratio, *Rr*: 9.9%; irreversible fouling ratio, *Rir*: 0.9%) and 43.4% at 0 V (reversible fouling ratio, *Rr*: 6.2%; irreversible fouling ratio, *Rir*: 56.6%). Compared with other reported membranes for oil fouling control (Table 1, Supplementary Table 1), the as-prepared Mn/BaTiO$_3$ membrane exhibits enhanced anti-fouling performance.

Moreover, to further validate the long-term self-cleaning effect of the membrane on typical oil fouling, the test duration was extended to 900 min. Even when challenged with a high-concentration emulsion of 2500 ppm, the membrane still maintained an anti-fouling efficiency of 75.7% (Supplementary Fig. 4). Considering factors such as membrane lifespan, anti-fouling efficiency, retention rate, and the value of the applied voltage, the Mn/BaTiO$_3$ piezoelectric membrane demonstrates a clear superiority over other reported piezoelectric or advanced membranes (Table 1, Supplementary Table 1). In this study, the Mn/BaTiO$_3$ piezoelectric membrane demonstrates sustained anti-fouling performance at an optimal 20 V, surpassing traditional piezoelectric membranes that require 40 - 100 V for operation (Table 1). Given operational safety and energy saving concerns, an applied voltage of 20 V is considered practically appropriate.

To assess membrane performance in real wastewater treatment, anti-fouling tests were conducted on oily emulsion wastewater from the food industry. As illustrated in Supplementary Fig. 5, the average oil particle size in this real wastewater was 1.27 μm, with a Total Organic Carbon (TOC) content of 862 ppm, which is lower than 2500 ppm of oil emulsion prepared for testing. Under an AC stimulation of 20 V, 265 kHz, the membrane demonstrated a 10.9% reduction in flux over an extended 900 min test time, with an oil particle retention rate exceeding 99.9%. The Mn/BaTiO$_3$ piezoelectric membrane exhibited competitive practical application value (Table 1, Supplementary Table 1).

We utilized an O/W emulsion stained with the lipid-soluble Oil Red O dye to visually demonstrate the oil retention performance of the Mn/BaTiO$_3$ piezoelectric membranes. The results, as depicted in Fig. 3a–c,

showed that the filtrates were remarkably clear and transparent. Further analysis through the Fluorescence Excitation-Emission Matrices (FEEMs, Fig. 3d, f) revealed that the membrane pore physical size sieving effect effectively removed a majority of oil droplets during various filtration processes. Additionally, the TOC removal rates, which remained consistently above 98.4% after 180 min of filtration is exemplary, especially when compared to other membranes reported in the literature (Table 1, Supplementary Table 1), highlighting the superior performance of the Mn/BaTiO$_3$ piezoelectric membrane.

**Against various foulants**

Foulants in water include both organic and inorganic substances, with oil being just one typical example of organic foulants. The Mn/BaTiO$_3$ piezoelectric membrane not only excels in oil fouling control, outperforming other advanced anti-fouling membranes (as listed in Table 1, Supplementary Table 1), but also demonstrates remarkable versatility. To assess its anti-fouling capabilities for various foulants, the membrane was challenged with a series of representative micro-nanoscale foulants (Supplementary Fig. 6), including microbes such as *Escherichia coli* (*E. coli*, $<d>$ = 1.69 μm, ζ = −16.1 mV), and inorganic colloidal particulate matter in wastewater (2500 ppm) with different charges (Al$_2$O$_3$ $<d>$ = 0.42 μm, ζ = +39.2 mV) and SiO$_2$ ($<d>$ = 0.56 μm, ζ = −23.5 mV). In all cases, the Mn/BaTiO$_3$ membrane, under a stimulation of 20 V at 265 kHz, maintained over 75% of its initial flux after 180 min: 89.6% for SiO$_2$, 84.8% for Al$_2$O$_3$, and 75.8% for *E. coli* (Fig. 4a). This performance starkly contrasts with the control group without the piezoelectric effect (0 V), which showed significantly lower anti-fouling efficiencies of 32.0% for *E. coli*, 38.8% for Al$_2$O$_3$, and 42.7% for SiO$_2$ in the cross-flow filtration system (Fig. 4b). Furthermore, when tested with a mixed solution of these foulants (combined in equal volumetric ratios) to mimic complex wastewater conditions, the membrane showed only about a 20% flux reduction over 180 min (Fig. 4c). Electron microscopy results (Fig. 4d, Supplementary Figs. 7–9) and the elemental mapping of the post-filtration membrane surfaces (Figs. 2d, e, 5a) clearly illustrate the versatile anti-fouling effects (organic/inorganic) of the membranes under AC stimulation of 20 V, 265 kHz. Typically, the membrane maintained high anti-fouling

**Table 1 | Comparing Mn/BaTiO₃ membrane with traditional lead-based and other advanced lead-free piezoelectric water treatment membranes (taking typical oil fouling as an example)**

| Piezoelectric Membrane | Lead-containing | Applied voltage (V) | Flux (L m⁻²h⁻¹) | Oil rejection (%)/ concentration (ppm) | Anti-fouling efficiency (%)[a] | Single running time (min) | Reference |
|---|---|---|---|---|---|---|---|
| PZT | √ | 20 | ~85 | ~, 500 | ~85 | 180 | 14 |
| α-Al₂O₃/PZT | √ | 40 | 150 | 99.7, 2000 | 59 | 120 | 52 |
| PZT/Ti | √ | 60 | ~210 | ~95, 500 | 90–95 | ~120 | 53 |
| BaTiO₃/PVDF | × | 20 | ~58 | ~95, 397.3 | 85.9 | 180 | 54 |
| SiO₂-Al₂O₃-MgO | × | 60 | 272 | 99.7, 500 | ~85 | ~120 | 55 |
| ZrO₂-SiO₂ | × | 100 | ~200 | 97.5, 500 | ~75–85 | ~120 | 56 |
| Al₂O₃/α-quartz | × | 100 | 190 | 97.9, 500 | ~55 | 120 | 31 |
| Mn/BaTiO₃ | × | 20 | ~91 | 98.4, 2500 | 91 | 180 | This work |
| | | | | | 75.7 | 900 | |

[a]Only single-run anti-fouling test results are considered, excluding scenarios where membranes are cleaned and retested after fouling.

efficiency even against microorganisms such as *E. coli*, which can strongly adhere to membrane surface by their viscous extracellular polymers, as demonstrated in Fig. 4d and Supplementary Fig. 9[36,37].

The Mn/BaTiO₃ piezoelectric membrane, when stimulated with AC voltage, demonstrates a remarkably versatile anti-fouling capabilities, effectively repelling a wide range of foulants irrespective of their type or surface characteristics. However, it remains to be elucidated whether these versatile anti-fouling effects are exclusively attributed to the membrane built-in ultrasonic vibrations. Therefore, a comprehensive investigation into the anti-fouling mechanisms responsible for various foulants of Mn/BaTiO₃ piezoelectric membrane is crucial and merits an in-depth discussion.

### Anti-fouling mechanism insights

Membrane fouling is usually related to the physicochemical characteristics of both the membranes and the foulants, encompassing aspects such as surface hydrophobicity, electrostatic potential, and the size of pores and particles[38–40]. When stimulated with AC, the Mn/BaTiO₃ piezoelectric membrane demonstrated a remarkable ability to mitigate the deposition of colloids with opposite charges (Figs. 4a, 5a, Supplementary Figs. 7–8). This suggests that despite the presence of electrostatic interactions, the piezoelectric vibration predominantly drives the membrane's self-cleaning efficacy[15,22,41]. To verify the anti-fouling mechanism hypothesis, finite element method simulations were conducted, elucidating the correlation between membrane microscale displacement and applied AC conditions.

Finite element simulations reveal and quantify how varying AC frequencies and voltages trigger the membrane's microscale piezoelectric vibrations, illuminating the understanding of the anti-fouling mechanism[42,43]. As depicted in Fig. 5b and Supplementary Fig. 10, the application of AC ranging from 5 ~ 20 V at frequencies of 5 ~ 600 kHz initiates vibration of the Mn/BaTiO₃ piezoelectric membrane through the inverse piezoelectric effect. Typically, piezoelectric materials exhibit optimal vibration performance at their specific resonant frequencies[44,45]. In this case, the Mn/BaTiO₃ membrane, known for its high piezoelectric properties, demonstrated its optimal resonant frequency around 265 kHz within the tested range. Notably, the membrane's total spatial displacement increases with rising AC voltage, peaking at 461 nm at 20 V[8]. These simulation results align closely with the anti-fouling experiments (Figs. 2, 4), indicating that the built-in piezoelectric vibration on the membrane surface and within its pores plays a crucial role in actively preventing the adherence and accumulation of foulants, thereby ensuring effective and continuous anti-fouling performance.

It has been reported that high-frequency ultrasound over 100 kHz could trigger acoustic cavitation, leading to the creation of collapsing microbubbles in water[30,46,47]. This phenomenon results in exceptionally high temperatures and pressures in supercritical areas, and the collapse of these microbubbles produces ROS with potent oxidative capabilities[30,46,47]. Significantly, and often neglected in prior research on piezoelectric water treatment ceramic membranes, is the capability of membrane built-in piezoelectric ultrasonic vibration to in-situ generate ROS, and the subsequent role of ROS in mitigating membrane fouling[3,14,18]. A potential anti-fouling mechanism involves ROS-mediated oxidative disruption of the adhesion between colloidal foulants and the membrane surface, particularly relevant for organic foulants such as oil[3]. As shown in Fig. 5b, c and Supplementary Fig. 11, upon stimulation at the optimal resonant high frequency (265 kHz, 20 V), significant ROS signals of ·OH, $^1O_2$ and $H_2O_2$ were detected via the membrane built-in ultrasonic vibration. These ROS have redox potentials of 1.9 ~ 2.7 V for ·OH and 2.2 V for $^1O_2$ versus the standard hydrogen electrode, sufficient to oxidize the organic foulants used in this study, thereby confirming the aforementioned hypothesis[48]. Additionally, under the same conditions, no such phenomenon was observed in the control group of common $Al_2O_3$ ceramic membrane

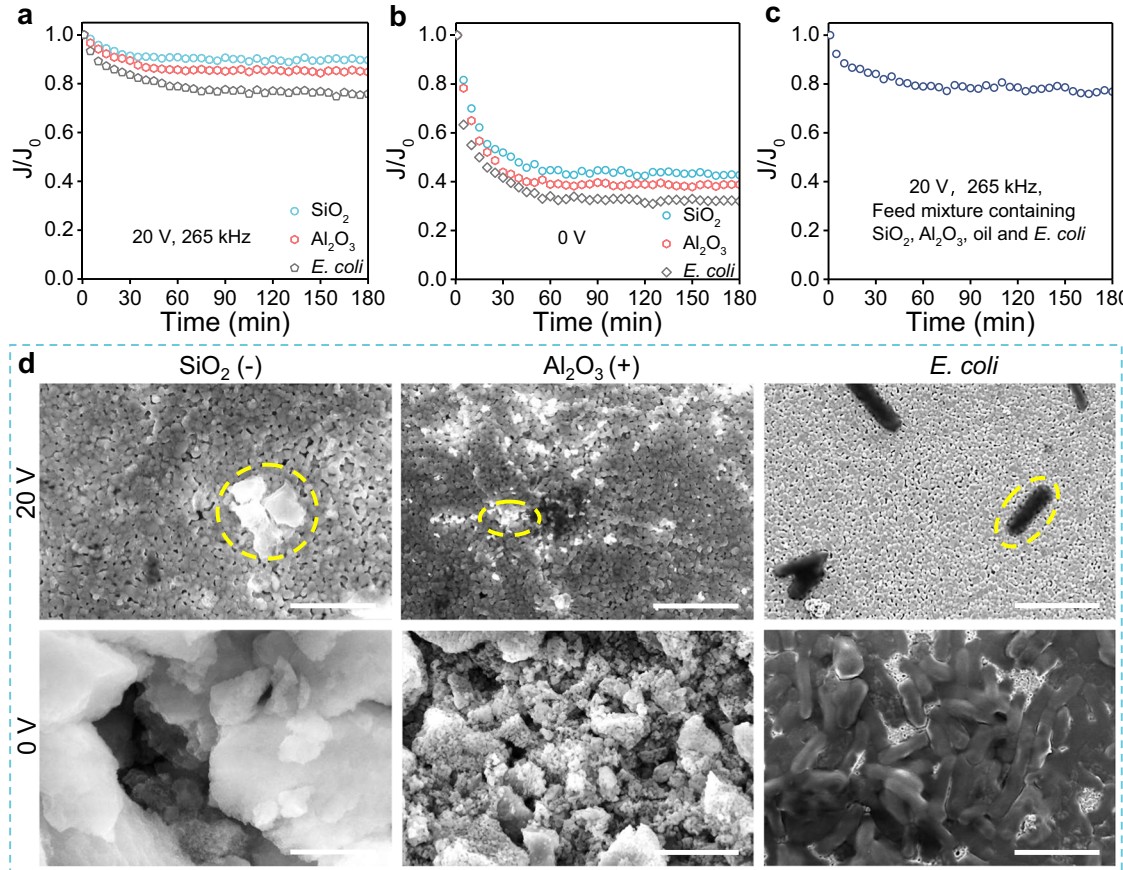

**Fig. 4 | Effects of the Mn/BaTiO₃ piezoelectric membrane against different foulants. a–c** Normalized membrane flux trends with various AC conditions for typical membrane foulants. **d** ESEM images of foulant-impacted membranes at different AC voltages (265 kHz), scale bar, 5 μm, representative foulants are delineated with yellow dashed lines for clarity.

without piezoelectric effect, eliminating the main possibility of ROS generation by AC alone (Fig. 5c).

The revelation that built-in piezoelectric ultrasonic vibration of the membrane leads to ROS generation, thereby facilitating the oxidation of organic foulants and contributing to their fouling control, represents a relatively novel phenomenon. The ROS quenching results presented in Fig. 5d indicate that the individual contributions of $^1O_2$ and ·OH to membrane fouling control in the oil fouling system are relatively minimal. Within the 180 min test duration, they account for only 4% and 9.6% of the overall fouling control, respectively. However, a significant decline in membrane flux is observed when both types of ROS are simultaneously quenched, showing a 17.5% of flux decrease compared to the group without ROS quenching, accounting for 19.8% of the total anti-fouling effect. This highlights the synergistic contribution of different ROS oxidation in anti-organic fouling. In comparison to the 0 V control group, which is devoid of membrane piezoelectric ultrasonic vibration and piezoelectric ROS generation, the removal of ROS in this system indicates that the contribution of membrane built-in piezoelectric ultrasonic vibration to anti-fouling is significant, accounting for 49.9%. Notably, the quantitative results detailing the contributions of each factor to anti-organic fouling should be viewed as referential, given that the synergy between the built-in piezoelectric ultrasonic vibration of the membrane and the ROS oxidation processes (Figs. 5d, 1b) enhances their collective efficacy. However, for the inorganic fouling, near-electrically neutral colloidal particles (SiO₂-hexadecyl trimethyl ammonium bromide, $\zeta = -0.27$ mV, Fig.5e) were used here to eliminate the influence of

surface charge on the particles. It was observed that the ROS oxidative anti-fouling effect for inorganic SiO₂ is limited, which primarily relies on the piezoelectric vibration of the Mn/BaTiO₃ membrane (Fig. 5e). Inorganic colloidal foulants, lacking the adhesive properties of organic foulants such as oil and microbes, merely deposit on membrane surface. Such deposits can be effectively removed through continuous membrane in-situ ultrasonic vibration, without requiring ROS oxidation for detachment[3,8,43]. These results could provide universal guidance for enhancing the understanding of the anti-fouling properties of piezoceramic membranes, which rely on membrane built-in piezoelectric ultrasonic vibration.

## Discussion

The development and application of the Mn/BaTiO₃ piezoceramic membrane, as detailed in this study, present significant environmental implications. When activated by AC, this membrane exhibits a significant enhancement in oil anti-fouling efficiency, with its performance increasing from 38.3% to 91.0% as the AC voltage is raised from 0 to 20 V. Furthermore, it delivers versatile anti-fouling performance regardless of the foulants' properties. Its self-cleaning property reduces the reliance on chemical cleaning agents, which can be environmentally harmful. Additionally, its enhanced water treatment efficiency, combined with a competitive cost compared to other reported piezoelectric or advanced membranes, is vital for sustainable development (Table 1, Supplementary Table 1)[49,50]. Finite element simulations not only optimize the operational parameters of the membrane but also elucidate the anti-fouling mechanisms at the

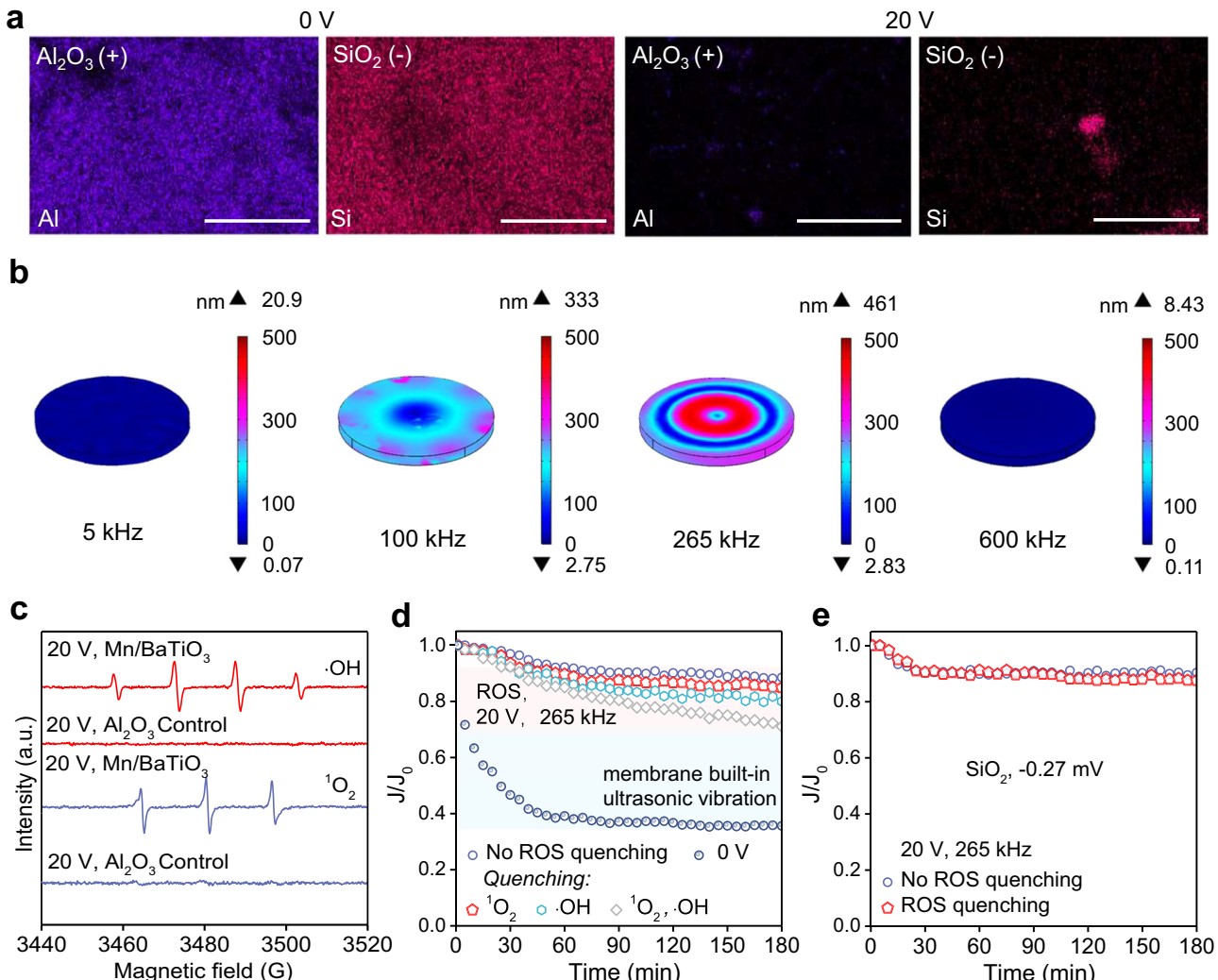

**Fig. 5 | Insights into the anti-fouling mechanism of Mn/BaTiO₃ piezoelectric membrane: synergistic effects of membrane built-in ultrasonic vibration and in-situ ROS generation. a** Elemental mapping of SiO₂ and Al₂O₃ on the surface of the Mn/BaTiO₃ membrane post-filtration. **b** Finite element simulations of membrane displacement at 20 V across different AC frequencies. **c** ROS detection during membrane processes, 20 V, 265 kHz. **d** Distinct contributions of ROS and membrane built-in piezoelectric ultrasonic vibration to anti-organic (oil) fouling. **e** The impact of ROS generated by Mn/BaTiO₃ piezoelectric membrane on anti-inorganic (near-neutral charge SiO₂ particles) fouling.

micro-scale level of the membrane. More importantly, considering this piezoceramic membrane as a case study, supplements and unifies the traditional anti-fouling mechanism, which previously relied solely on the built-in ultrasonic vibration of piezoceramic membranes[15,18]. It reveals that the membranes in-situ ultrasonic vibration could generate previously neglected ROS around the membranes, thereby offering potential synergistic benefits to the anti-fouling process[30,46,47]. This study offers a comprehensive self-cleaning solution and universal mechanism insights among piezoceramic membranes. Future studies could explore fabricating lead-free barium titanate piezoceramic ultrafiltration and nanofiltration ceramic membranes. This approach includes developing barium titanate piezoelectric thin layers on macro-porous supports to improve membrane flux, thereby increasing their applicability and scalability in water treatment. Additionally, by co-sintering functional catalysts with piezoelectric ceramics to form piezoelectric catalytic membranes, this approach not only combats membrane fouling but also advances catalytic membrane technology for wastewater treatment, with a particular focus on the treatment of persistent organic wastewaters.

## Methods

### Materials

Titanium dioxide (TiO₂, 99%), barium carbonate (BaCO₃, 99%), polyvinyl alcohol (PVA, alcoholysis degree, 98 - 99%), sodium dodecyl sulfate (SDS, C₁₂H₂₅SO₄Na, 99%), hexadecyl trimethyl ammonium bromide (CTAB, C₁₉H₄₂BrN, 99%), 1,2-dichloroethane (C₂H₄Cl, 99%), soybean oil (reagent grade), petroleum ether (water free), Oil Red O (C₂₆H₂₄N₄O, for biochemical research), aluminum oxide (Al₂O₃, 99.9%), 2,2,6,6-tetramethyl-4-piperidinol (TEMP), tertbutyl alcohol (TBA, C₄H₁₀O, 99%), β-carotene (C₄₀H₅₆, 96%), acetone and manganese oxide (Mn₂O₃, 99.9%) were obtained from Aladdin. 5,5-dimethyl-1-pyrolin-N-oxide (DMPO) was purchased from Dojindo Laboratories. The commercial H₂O₂ test kit was purchased from Solarbio. Silicon dioxide (SiO₂, 99.9%) was purchased from Mackin. Luria-Bertani (LB) was procured from Hopebiol. *Escherichia coli* (*E. coli*, ATCC® 23716™) was selected as the experimental strain. The real oily wastewater was obtained from local food service industry. All experimental solutions were prepared with deionized water (18.2 MΩ·cm).

## Membrane fabrication

Synthesis of BaTiO$_3$ powders: Initially, BaCO$_3$ and TiO$_2$ were in an equimolar ratio and were mixed in alcohol, then ball-milled at 550 rpm for 16 h, followed by drying at room temperature. The mixed powders were then calcined in air at 1180 °C for 4 h, maintaining a steady heating rate of 5 °C min$^{-1}$. After calcination, the powders were ball-milled using the previously described method to obtain the final BaTiO$_3$ powders[32].

Sintering of the membranes: Powders of BaTiO$_3$ with 0.1% Mn$_2$O$_3$ were milled with ethanol by ball milling in a planetary ball mill (XGB-04, Boyuntong Instrument Technology, China) at 550 rpm for 12 h. This was followed by drying at 70 °C for 8 h, and then the mixture was sintered at 1000 °C for 4 h. The resulting sintered powders were milled and dried under the aforementioned conditions, after which an aqueous solution of PVA (8% w/w) was added, and the mixture was uniaxially compressed at 15 MPa for 30 s to shape circular membranes. Upon drying at room temperature overnight, these green-pressed membranes were then sintered in a high-temperature muffle furnace at a heating rate of 5 °C min$^{-1}$ and kept at 1180 °C for 4 h. Finally, the membranes were allowed to naturally cool to room temperature.

Poling of the membranes: The sintered membranes, equipped with copper electrodes on both sides, were submerged in a paraffin bath heated to 110 °C. A poling treatment, applying a direct-current (DC) voltage of 3 kV mm$^{-1}$, was conducted for 1 h. After poling, the membranes were immersed in ethanol at 70 °C for an hour and rinsed three times with ultrapure water to eliminate all traces of paraffin and ethanol, and ultimately obtained the piezoelectric Mn/BaTiO$_3$ membranes.

## Characterization

The surface and cross-sectional morphology of the membranes were examined using Environmental Scanning Electron Microscopy (ESEM, Quanta FEG-250), complemented by energy dispersive X-ray Spectroscopy (EDS) for detailed elemental analysis. Additionally, Scanning Transmission Electron Microscopy (STEM) and High-Angle Annular Dark Field (HAADF) imaging, paired with EDS mapping, were employed to assess the elemental distribution of the Mn/BaTiO$_3$ grains using FEI, TF20. X-ray Diffraction (XRD) patterns were captured using a Bruker D8 Advance diffractometer, employing Cu Kα radiation to scan a 2θ angle range from 20° to 80° with a step size of 0.05°. Fluorescence Excitation-Emission Matrices (FEEMs) of the samples were recorded with a Hitachi F7000 fluorescence spectrophotometer. The Total Organic Carbon (TOC) content in the feed solution and filtrate was analyzed using a Shimadzu TOC-L analyzer. Particle size distribution, mean particle size, and zeta potentials were determined using a Malvern Zeta-sizer Nano ZS90 laser scattering analyzer. The pore size distribution of the sintered membrane was calculated using Nano Measurer 1.2.0 software, based on the ESEM image of Fig. 1b (surface, scale bar 5 μm). Membrane porosity was calculated using the gravimetric method defined as: porosity (%) = 100 × (G$_3$ − G$_1$)/(G$_3$ − G$_2$), where G$_3$ represents the weight of the thoroughly water-wetted membrane, G$_1$ is the dry weight of the membranes, and G$_2$ is the submerged weight determined by water buoyancy. The underwater oil contact angle for a 10 μL droplet of 1,2-dichloroethane on the membrane surface was measured using a DSA100 video-based contact angle measuring device from Kruss Scientific. The total spatial displacement of the piezoelectric membranes under various AC conditions was simulated using COMSOL Multiphysics® 5.6, establishing the physical field of the membranes in a three-dimensional domain.

## Preparation of the O/W emulsion wastewater

An O/W emulsion was prepared by adding the appropriate weight of soybean oil, sodium dodecyl sulfate, and Oil Red O dye to deionized water. This mixture was subjected to ultrasonic dispersion at room temperature for 1 h, followed by mechanical agitation for 12 h.

## Preparation of the *E. coli* suspension

25 g of Luria-Bertani medium was dissolved in 0.95 L of deionized water, and the solution was then autoclaved at 121 °C for 20 min to sterilize. After cooling, 1 mL of culture was added to the sterilized medium and incubated at 37 °C, 160 rpm for 48 h. The *E. coli* feed solution was prepared by diluting the liquid culture medium tenfold to simulate the real microbial wastewater, which typically contains a mixture of nutrients and microorganisms.

## Detection of ROS

The Electron Paramagnetic Resonance (EPR) spectra of 5,5-dimethyl-pyrroline N-oxide (DMPO)-·OH and 2,2,6,6-tetramethylpiperidine (TEMP)-$^1$O$_2$ were recorded using an EMX-10/12 (Bruker) to detect the generation of ROS induced by the built-in ultrasonic vibration of different membranes, under the application of a 20 V AC voltage. 200 mL of deionized water with 22 mL of 1 mmol L$^{-1}$ DMPO or 200 mL of deionized water with 50 mL of 0.5 mmol L$^{-1}$ TEMP for ·OH and $^1$O$_2$ detection, respectively. Different radical scavengers were introduced into the reaction system, including TBA (scavenger for ·OH) and β-carotene (scavenger for $^1$O$_2$, acetone as solvent)[51]. The Mn/BaTiO$_3$ piezoelectric membrane was submerged in 100 mL of deionized water. After applying various AC voltages to its surface for 5 min, water samples were collected for further analysis to detect generated H$_2$O$_2$. Detection was performed using a commercial test kit, according to the procedures specified in the manual.

## Membrane performances

The filtration and anti-fouling capabilities of the membranes were assessed using a lab-scale cross-flow filtration system operated at a pressure of 1 bar. The system featured a cross-flow membrane module, outfitted with two porous steel electrodes for applying AC signals (UTG9003C, UNIT Inc. China) to the Mn/BaTiO$_3$ piezoelectric membranes. During the separation process, constant stirring was maintained to ensure the stability of the feed solution. The filtrate volume was monitored and recorded at 5 min intervals, and the permeate was collected in a reservoir situated on an electronic balance. The membrane flux (*J*) was calculated over time using the specified equation Eq. (1):

$$J = \frac{V}{A \times \Delta t} \tag{1}$$

where V (L), A (m$^2$) and Δt (h) denote the volume of permeation, the effective membrane area, and the testing duration, respectively. Membrane fouling was assessed by tracking the normalized membrane flux (*J/J$_o$*) over time, where *J$_o$* represents the initial membrane flux. The oil rejection *R$_{TOC}$* (%) was calculated according to Eq. (2):

$$R_{TOC}(\%) = \frac{C_f - C_p}{C_f} \times 100 \tag{2}$$

where C$_f$ and C$_p$ represent the TOC concentrations of the feed and permeate, respectively.

To evaluate the synergy between the built-in piezoelectric ultrasonic vibration and in-situ ROS on membrane fouling, tests were conducted on membranes both with and without the application of AC voltage. The flux recovery ratio (*FRR*), reversible fouling ratio (*Rr*), and irreversible fouling ratio (*Rir*) were utilized to assess the anti-fouling

performance and fouling resistance as follows[8]:

$$FRR(\%) = \frac{q_2}{q_0} \times 100 \qquad (3)$$

$$R_r(\%) = \frac{q_2 - q_1}{q_0} \times 100 \qquad (4)$$

$$R_{ir}(\%) = \frac{q_0 - q_2}{q_0} \times 100 \qquad (5)$$

Where $q_O$ is the pure water flux, $q_1$ is the flux of O/W emulsion of 2500 ppm, and $q_2$ is the pure water flux the membrane after cleaning (membranes were cleaned by rinsing with pure water for 20 min, L m$^{-2}$ h$^{-1}$ bar$^{-1}$).

## Reporting summary
Further information on research design is available in the Nature Portfolio Reporting Summary linked to this article.

## Data availability
The data supporting the findings of the study are included in the main text and supplementary information files. Raw data can be obtained from the corresponding author upon request.

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

## Acknowledgements

This work is supported by the National Natural Science Foundation of China (52371346 (Z.Y.), 22306026 (Z.Y.)), Young Elite Scientists Sponsorship Program by China Association for Science and Technology (2023QNRC001 (Z.Y.)), Ecological Society of China (STQT2023C07 (Z.Y.)), the Fundamental Research Funds for the Central Universities (2242024K40007 (Z.Y.)), and the Start-up Research Fund of Southeast University (RF1028623141 (Z.Y.)).

## Author contributions

Y.Z. conceived the idea, carried out the experiments and finite element simulation, and wrote the paper. F.Y., H.J., and G.D.G. helped with data analysis and manuscript polishing. All the authors discussed results and provided comments during the manuscript preparation.

## Competing interests

The authors declare no competing interests.
