## [Peer Review File · Nature Communications]

Piezoceramic membrane with built-in ultrasound for reactive oxygen species generation and synergistic vibration anti-foulingREVIEWER COMMENTS

Reviewer #2 (Remarks to the Author):

In this work, authors reported a Mn/BaTiO₃ piezoceramic membrane and revealed its anti-fouling mechanism during wastewater treatment. The study in this manuscript is systematic but the innovation is still insufficient. So, I think this manuscript is not appropriate to be published in Nature Communications. Major comments are listed below.

- (1) In the author's previous manuscript (Nature, 2022, 608, 69-73), the MnO/BaTiO₃ piezoelectric membrane was reported. I found the formulation of Mn/BaTiO₃ piezoceramic membrane in this manuscript is same as the previous work. The innovation of this work should be clarified.
- (2) The Mn/BaTiO₃ piezoceramic membrane in this work with pore size of 210 nm and water permeability of 91 LMH·bar⁻¹. The performance, especially permeability, of this membrane is not advantage to most membranes with similar pore size.
- (3) What's the power of the piezoelectric membrane in this work? Cavitation effects may not occur at low power.
- (4) There are still some mistakes in this manuscript. For example, line 254, first line in page 15.

Reviewer #3 (Remarks to the Author):

In this manuscript, the authors propose a synergistic anti-fouling strategy through the in-situ vibration of piezoelectric membrane and the generation of ROS. Membrane fouling is indeed an unsolved problem in membrane technology. Piezoelectric water treatment membranes exhibit great potential in membrane fouling control, but these membrane materials may be dependent on lead or exhibit low piezoelectric performance. Furthermore, the anti-fouling mechanism of both organic and inorganic piezoelectric membranes is currently thought to be due to the singular effect of piezoelectric vibration. In this manuscript, the authors introduce a lead-free, high piezoelectric performance BaTiO₃ based piezoelectric ceramic membrane. This membrane effectively against multiple typical pollutants in wastewater and demonstrate good practical application value with actual oily wastewater. The authors have compared this membrane to existing piezoelectric membranes, and it appears to be more competitive in terms of anti-fouling performance and energy efficiency. More importantly, this original work unveils the deep mechanism behind the anti-fouling of piezoelectric membranes, namely, that in-situ piezoelectric vibration can generate ROS to synergistically combat fouling. This scientific discovery can provide technical support for the development of piezoelectric water treatment membranes. The ROS generated in-situ by the piezoelectric membrane not only aids in the anti-fouling of piezoelectric membranes but can also be extended to other areas such as water treatment catalytic membranes, for the degradation or removal of other water pollutants. This manuscript is mostly written clearly, and the results are well

presented in figures of good quality. Even though the results are encouraging, my concerns are :1) In the introduction, the authors discuss the disadvantages of organic piezoelectric membrane materials compared to piezoelectric ceramic materials, such as the lower piezoelectric coefficient of PVDF, which makes it more susceptible to contamination. However, as far as we know, potentially more cost-effective organic piezoelectric materials like PTFE, which have excellent piezoelectric catalytic effects, could they potentially replace piezoelectric ceramics in anti-membrane fouling applications?

2) At line 130, the best performance was achieved at the highest tested voltage (20V). Would performance improve under higher voltages?

3) In Figure 2, how does the frequency of the alternating current affect the anti-fouling effect, and why doesn't the anti-fouling efficiency increase proportionally with the increase in AC frequency?

4) Is there a possibility of co-sintering the BaTiO₃ piezoelectric ceramic membrane with other inorganic catalysts during the sintering process? This requires further explanation.

5) At line 258, the generation of ROS is accompanied by the production of different free radicals; is there a possibility that H₂O₂ is produced in this process? The authors should conduct a deeper investigation into this mechanism.

Response to reviewers' comments and changes made in the manuscript

Manuscript ID: NCOMMS-24-01399A

Piezoceramic membrane with built-in ultrasound for reactive oxygen species generation and synergistic vibration anti-fouling

Reviewer #2

General Comment: In this work, authors reported a Mn/BaTiO₃ piezoceramic membrane and revealed its anti-fouling mechanism during wastewater treatment. The study in this manuscript is systematic but the innovation is still insufficient. So, I think this manuscript is no appropriate to be published in Nature Communications. Major comments are listed below.

Response: Thank you for your constructive comments aimed at enhancing the quality of our paper, as well as the positive assessment of the systematic nature of our work. From your detailed comments, we sense there might be a significant misunderstanding regarding our work, particularly concerning its innovativeness. We apologize for any confusion caused by the lack of clarity in our writing and request the opportunity to clarify. Through detailed revisions to our manuscript (with additions or changes highlighted in blue), we hope to accurately convey the core focus and innovative aspects of our work.

Firstly, the piezoceramic membrane used in this work outperforms the currently reported piezoceramic membranes in terms of energy saving and anti-fouling effectiveness and universality (as shown in Tables S1-2). More importantly, the essence of this study is to demonstrate, through the use of this piezoceramic membrane, that piezoelectric vibrations in membrane processes can generate reactive oxygen species (ROS), and to elucidate the synergistic mechanism of ROS with the universality of piezoelectric vibrations in enhancing the anti-fouling properties of piezoceramic membranes (as indicated in the title), rather than merely introducing a high-performance piezoceramic anti-fouling water treatment membrane. It should be noted that, since at least 1992, the mechanism of anti-fouling has been solely

attributed to the physical vibration of the piezoceramic membrane, which has clearly hindered further development and application. From the perspective of the mechanisms, this work represents a significant breakthrough in the application of piezoceramic membranes for water treatment and anti-fouling. Moreover, starting from anti-fouling, the proposed ROS mechanism also serves as a reference for the development of piezoceramic catalytic membranes for water treatment, appealing to a broad audience. We sincerely thank you again for your constructive comments on improving the quality of our paper, with updates reflected in the detailed responses below.

Comment 1: In the author's previous manuscript (Nature, 2022, 608, 69-73), the MnO/BaTiO₃ piezoelectric membrane was reported. I found the formulation of Mn/BaTiO₃ piezoceramic membrane in this manuscript is same as the previous work. The innovation of this work should be clarified.

Response: Thank you for your attention to our previous work, which indeed requires clarification to distinguish between the two studies and to highlight the innovation of this paper. It is important to note that the piezoelectric effect is divided into the direct piezoelectric effect (where mechanical stress is converted into electrical charge) and the inverse piezoelectric effect (where electrical voltage is converted into mechanical stress or vibration). Both effects are equally important and collectively referred to as the piezoelectric effect.

In our work published in Nature, we introduced the concept of applying the direct piezoelectric effect to combat membrane fouling: through hydraulic pressure-driven membrane processes, the pulse hydraulic pressure activates the piezoceramic membrane to generate a piezoelectric electric field and create dielectrophoretic effects for anti-fouling among other piezoelectric activities. Taking anti-fouling as an example, we broadly extended this principle to the anti-fouling effects of other materials under scenarios involving hydraulic fluctuations, such as the anti-fouling of ship surfaces with piezoelectric coatings under ocean wave impacts, or domestic ceramic toilets made from piezoceramic that combat fouling through the direct piezoelectric effect during water flushing. The core idea we wish to convey is leveraging environmental pressure conditions to stimulate piezoelectric materials into generating electroactivity for

anti-fouling. The specific piezoceramic materials or anti-fouling applications we describe are merely the mediums or tools for expressing this core concept, as our work aims for universality, not restricted to the piezoelectric materials reported.

Regarding this work, in addition to our previously reported work in Nature on the direct piezoelectric effect for anti-fouling, the use of the piezoelectric effect for anti-fouling since at least 1992 has been based on the inverse piezoelectric effect. This involves stimulating piezoceramic membrane materials with alternating current to convert electrical energy into mechanical energy (the piezoelectric ultrasonic vibration of the membrane material) for anti-fouling. Moreover, for decades, the mechanism was singularly attributed to the anti-fouling effect of vibration without any development. Our work, utilizing the piezoceramic membrane from our previous Nature publication as a medium, explains how the inverse piezoelectric effect can generate reactive oxygen species (ROS) through the membrane's own piezoelectric ultrasonic vibrations and synergize with piezoelectric vibrations for anti-fouling, as indicated in the title of this paper. Specifically, for certain organic pollutants, the oxidative action of ROS plays a crucial role in effectively combating these contaminants, a mechanism previously unrecognized. By demonstrating that the inverse piezoelectric membrane process can generate ROS, our work also positively impacts the development of piezoelectric catalytic membranes, appealing to a broad audience. Similarly, this idea or mechanism discovery is the core of our work; the specific piezoceramic material is merely a tool to express this idea, as our work provides a universal understanding of the mechanism behind piezoceramic membranes. Of course, publications in high-quality journals have high standards in all respects, including the anti-fouling performance of the membrane. Therefore, as a continuation of our previous work, we are eager to demonstrate the application of this piezoceramic membrane in combating membrane fouling through the inverse piezoelectric effect in this current work. Through these studies, we have comprehensively presented our innovative ideas for using piezoelectric effects to control membrane fouling, addressing both direct and inverse effects, which are independent and equally important. We appreciate your valuable suggestion and have enhanced our explanation in this regard; the relevant content has been added to the manuscript.

Line 23-26: This offers profound insights into the distinct anti-fouling processes for organic and

inorganic wastewater, supplementing and unifying the traditional singular vibrational anti-fouling mechanism of piezoceramic membranes, and potentially propelling the development of piezoelectric catalytic membranes.

Line41-48: Active anti-fouling strategies, especially those utilizing continuous ultrasonic vibration or photocatalytic activities, have demonstrated notable effectiveness^{10, 11, 12}. However, issues like membrane deterioration, the necessity for cavitation regulation, and high energy consumption still limit their practical applications^{8, 11}. Fortunately, these drawbacks of conventional ultrasonic approaches can be overcome through the use of self-vibrating piezoelectric membranes that harness the **inverse piezoelectric effect**^{13, 14}. The inverse piezoelectric effect, which converts Alternating Current (AC) stimulation into built-in membrane ultrasonic vibrations, has been conclusively shown to mitigate fouling, thus improving membrane flux and extending its operational life^{13, 14}.

Line69-80: In prior research, we harnessed the inherent hydraulic pressure of hydraulically driven membrane processes to elicit **the direct piezoelectric effect** in the Mn-doped BaTiO₃ (Mn/BaTiO₃) piezoceramic membrane, thereby controlling fouling³. This piezoceramic membrane transforms periodic pressure fluctuations into electrical pulses and rapid voltage changes, thus generating reactive oxygen species (ROS) and dielectrophoretic (DEP) forces at the membrane surface. ROS degrades foulants via oxidation or severs their attachment to the membrane surface, followed by DEP force driving the foulants away from the membrane surface. Since DEP force is independent of the type of foulants, this mechanism offers a universal approach to fouling control³. Notably, the inverse piezoelectric effect, corresponding to the direct piezoelectric effect, is also crucial in anti-fouling applications via piezoelectric ultrasonic vibration. Since at least 1992, there have been reports of using piezoelectric ultrasonic vibration for membrane fouling mitigation. Despite decades of research, this remains the sole mechanism identified, lacking further breakthroughs^{23, 28, 29}.

Line86-91: In this study, through the Mn/BaTiO₃ piezoceramic membrane, we comprehensively discussed the anti-fouling performances and potential mechanism insights of piezoceramic membranes via the inverse piezoelectric effect, as illustrated in Fig. 1a-b. An in-depth analysis was conducted to determine the optimal operating conditions for the membrane. Additionally, the study investigated the

essential anti-fouling mechanisms of membranes, focusing on built-in piezoelectric ultrasonic vibration and ROS synergy, and combined theoretical simulations with experiments across diverse wastewater scenarios.

Line 300-303: Additionally, by co-sintering functional catalysts with piezoelectric ceramics to form piezoelectric catalytic membranes, this approach not only combats membrane fouling but also advances catalytic membrane technology for wastewater treatment, with a particular focus on the treatment of persistent organic wastewaters.

Comment 2: The Mn/BaTiO₃ piezoceramic membrane in this work with pore size of 210 nm and water permeability of 91 LMH·bar⁻¹. The performance, especially permeability, of this membrane is not advantage to most membranes with similar pore size.

Response: Thank you for your comments. As we all know, membrane flux is closely related not only to its pore size and porosity but also to membrane thickness. The membrane discussed in this paper is a ceramic membrane, with a thickness ranging between 2 and 2.2 mm. This thickness is significantly different from that of organic membranes with similar pore sizes, which typically have a thickness of about 0.1 mm, a difference of approximately 20 times. Therefore, the membrane flux can only be compared to other ceramic membranes. As shown in Table S2, compared to reported advanced ceramic/piezoceramic membranes, our ceramic membrane exhibits comparable flux while demonstrating superior anti-fouling performance. More importantly, as you should understand from the above response, the core of this paper is not to introduce a piezoceramic membrane with exceptional performance. One method to improve flux is to apply a thin layer of piezoceramic on the surface of other macroporous membrane substrates. Thank you again for your very representative comments; the related content has been updated in the main text:

Line 99-103: It possesses an average pore size of 210 nm (Supplementary Fig. 1b), a bulk porosity of 22.43%, and measures 30 mm in diameter with a thickness ranging from 2 to 2.2 mm. This membrane exhibits hydrophilic properties, as indicated by an underwater oil angle exceeding 140° (Supplementary

Fig. 1c-d) and a pure water permeability of ~ 91 LMH bar⁻¹, which correlates with membrane thickness.

Line 296-300: Future studies could explore fabricating lead-free barium titanate piezoceramic ultrafiltration and nanofiltration ceramic membranes. This approach includes developing barium titanate piezoelectric thin layers on macro-porous supports to improve membrane flux, thereby increasing their applicability and scalability in water treatment.

Supplementary Table 2. Comparing Mn/BaTiO₃ membrane with traditional lead-based and other advanced lead-free piezoelectric water treatment membranes (taking typical oil fouling as an example).

Piezoelectric Membrane	Lead-containing	Applied voltage (V)	Flux (L m ⁻² h ⁻¹)	Oil rejection (%) / concentration (ppm)	Anti-fouling efficiency (%)*	Single running time (min)	Reference
PZT	√	20	~85	—, 500	~85	180	8
α-Al ₂ O ₃ /PZT	√	40	150	99.7, 2000	59	120	9
PZT/Ti	√	60	~210	~95, 500	90-95	~120	10
BaTiO ₃ /PVDF	×	20	~58	~95, 397.3	85.9	180	11
SiO ₂ -Al ₂ O ₃ -MgO	×	60	272	99.7, 500	~85	~120	12
ZrO ₂ -SiO ₂	×	100	~200	97.5, 500	~75-85	~120	13
Al ₂ O ₃ /α-quartz	×	100	190	97.9, 500	~55	120	14
Mn/BaTiO ₃	×	20	~91	98.4, 2500	91 75.7	180 900	This work

* Only single-run anti-fouling test results are considered, excluding scenarios where membranes are cleaned and retested after fouling.

Comment 3: What's the power of the piezoelectric membrane in this work? Cavitation effects may not occur at low power.

Response: Thank you for your constructive suggestion. The vibration of piezoceramic can indeed facilitate ultrasonic cavitation, as the principle of ultrasound utilizes the core components of piezoceramic. Therefore, the new mechanism we propose for generating reactive oxygen species (ROS) is applicable to

piezoceramic membranes universally.

The formation of cavitation is related to multiple factors such as frequency, power, temperature, and the liquid environment, making it quite complex. To verify that low power can also generate cavitation at ultrasonic frequencies, we demonstrated with commercially mature ultrasonics at even lower power levels. (Fig. 1). The power of this handheld ultrasonic instrument is only 0.10 W/cm^2 , yet it can continuously generate noticeable cavitation bubbles in dye wastewater. After measurement and calculation, our membrane surface current is over 40 mA, and the power calculated based on voltage and effective membrane area is about 0.73 W/cm^2 , therefore, it meets the conditions for generating cavitation. Moreover, low power, from a lateral perspective, is conducive to energy saving.

Fig. 1 | Example of a low-power ultrasonic instrument and the cavitation it generates.

If you are referring to the magnitude of the voltage, we believe that the piezoceramic membrane discussed in this article not only excels in anti-fouling performance but also operates at the lowest possible voltage, as shown in Table S2. Based on your suggestion, we have added further explanation to the text.

Line 81-85: Actually, from an ultrasonic perspective, the core component of an ultrasonic generator is the piezoelectric ceramic or piezoelectric crystal, and the built-in ultrasonic resonant frequency of piezoceramic membranes usually falls in the high-frequency ultrasonic range ($>100 \text{ kHz}$). This high-

frequency can generate significant ROS for advanced oxidation processes in water treatment, which previous studies overlooked but is potentially crucial for fouling control in piezoceramic membranes^{3, 14, 30, 31}.

Line 154-158: In this study, the Mn/BaTiO₃ piezoelectric membrane demonstrates sustained anti-fouling performance at an optimal 20 V, surpassing traditional piezoelectric membranes that require 40~100 V for operation (Supplementary Table 2). Given operational safety and energy saving concerns, an applied voltage of 20 V is considered practically appropriate.

Comment 4: There are still some mistakes in this manuscript. For example, line 254, first line in page 15.

Response: Thank you for your meticulous review and efforts. We have not only corrected the two typographical errors you pointed out but also conducted a comprehensive error check throughout the manuscript. We hope our sincere response will provide you with a deeper understanding and appreciation of our work. Once again, we genuinely thank you for your constructive suggestions that have helped improve the quality of our work.

Reviewer #3

General Comment: In this manuscript, the authors propose a synergistic anti-fouling strategy through the in-situ vibration of piezoelectric membrane and the generation of ROS. Membrane fouling is indeed an unsolved problem in membrane technology. Piezoelectric water treatment membranes exhibit great potential in membrane fouling control, but these membrane materials may be dependent on lead or exhibit low piezoelectric performance. Furthermore, the anti-fouling mechanism of both organic and inorganic piezoelectric membranes is currently thought to be due to the singular effect of piezoelectric vibration. In this manuscript, the authors introduce a lead-free, high piezoelectric performance BaTiO₃ based piezoelectric ceramic membrane. This membrane effectively against multiple typical pollutants in wastewater and demonstrate good practical application value with actual oily wastewater. The authors have compared this membrane to existing piezoelectric membranes, and it appears to be more competitive in terms of anti-fouling performance and energy efficiency. More importantly, this original work unveils the deep mechanism behind the anti-fouling of piezoelectric membranes, namely, that in-situ piezoelectric vibration can generate ROS to synergistically combat fouling. This scientific discovery can provide technical support for the development of piezoelectric water treatment membranes. The ROS generated in-situ by the piezoelectric membrane not only aids in the anti-fouling of piezoelectric membranes but can also be extended to other areas such as water treatment catalytic membranes, for the degradation or removal of other water pollutants. This manuscript is mostly written clearly, and the results are well presented in figures of good quality. Even though the results are encouraging, my concerns are:

Response: Thank you for your high appraisal of our work's quality and innovation, as well as for your diligent review efforts. Under your advisement and guidance, as you have noted, starting from anti-fouling, the introduction of the ROS mechanism within this process offers insights for the development of water treatment catalytic membranes, reaching a wide audience. We extend our gratitude once again for your contributions to enhancing the quality of the paper. The sections that have been added or altered have been distinguished with blue font in the manuscript.

Comment 1: In the introduction, the authors discuss the disadvantages of organic piezoelectric membrane materials compared to piezoelectric ceramic materials, such as the lower piezoelectric coefficient of PVDF, which makes it more susceptible to contamination. However, as far as we know, potentially more cost-effective organic piezoelectric materials like PTFE, which have excellent piezoelectric catalytic effects, could they potentially replace piezoelectric ceramics in anti-membrane fouling applications?

Response: Thank you for your comment, which opens up a broad area for discussion. Organic piezoelectric materials like PVDF and PTFE have a lower piezoelectric coefficient compared to piezoceramics and are more susceptible to contamination by oils and other pollutants relative to ceramic membranes. As you mentioned, although some studies have reported the efficient catalytic action of PTFE, its mechanism has been attributed to triboelectric or contact electrification, not piezoelectric effects (*Nat Commun* 13, 130 (2022); *Nat Commun* 15, 757 (2024)), which is unrelated to the focus of this paper. Furthermore, the high hydrophobicity of PTFE is not suitable for use as a water treatment membrane material, as it would significantly limit the membrane's flux. Therefore, organic materials like PTFE cannot replace piezoceramics in water treatment applications for anti-fouling. This content is discussed in the text as follows:

Line 49-55: Organic piezoelectric response membranes, notably polyvinylidene fluoride (PVDF) types, have been recognized for their hydrophilicity, cost-effectiveness, and efficient anti-fouling via built-in membrane ultrasonic vibration¹⁵. Their disadvantage lies in the typically low piezoelectric coefficient ($d_{33} \leq 30$ pC/N), limited durability to stubborn foulants like oil and bacteria^{16,17}. These limitations highlight the need for alternative piezo-materials that offer enhanced anti-fouling performances with durability^{14,18}. Ceramic membranes offer superior chemical stability and mechanical strength. Typically, the piezoceramics could self-clean by in-situ vibration under AC stimulation^{14,17}.

Comment 2: At line 130, the best performance was achieved at the highest tested voltage (20V). Would performance improve under higher voltages?

Response: Thank you for raising an interesting question, and I appreciate your comment. According to piezoelectric theory, piezoelectric vibrations correlate positively with the voltage applied; theoretically, a higher voltage should enhance the anti-fouling effect, which aligns with our experimental outcomes. However, considering that applying a 20 V voltage in our work achieved and even surpassed the anti-fouling performance reported for other piezoelectric membranes at 40 V, and even 100 V, securing over 90% long-term anti-fouling efficiency, opting for higher voltages to further improve anti-fouling effects might not be the best approach. This also brings into consideration the safety concerns associated with high-voltage operations. Therefore, we did not pursue higher voltages for further anti-fouling studies in this paper. The related content is as follows:

Line 128-131: Furthermore, increasing the intensity of AC, particularly at a resonance frequency of 265 kHz, amplified the vibrational amplitude of membrane. *As a result, when the AC voltage was raised from 0 to 20 V, there was a corresponding enhancement in anti-fouling performance, ranging from 38.3% to 91.0% (Fig. 2b-c).*

Line 154-158: *In this study, the Mn/BaTiO₃ piezoelectric membrane demonstrates sustained anti-fouling performance at an optimal 20 V, surpassing traditional piezoelectric membranes that require 40~100 V for operation (Supplementary Table 2). Given operational safety and energy saving concerns, an applied voltage of 20 V is considered practically appropriate.*

Supplementary Table 2. Comparing Mn/BaTiO₃ membrane with traditional lead-based and other advanced lead-free piezoelectric water treatment membranes (taking typical oil fouling as an example).

Piezoelectric Membrane	Lead-containing	Applied voltage (V)	Flux (L m ⁻² h ⁻¹)	Oil rejection (%) / concentration (ppm)	Anti-fouling efficiency (%)*	Single running time (min)	Reference
PZT	√	20	~85	—, 500	~85	180	8
α-Al ₂ O ₃ /PZT	√	40	150	99.7, 2000	59	120	9
PZT/Ti	√	60	~210	~95, 500	90-95	~120	10
BaTiO ₃ /PVDF	×	20	~58	~95, 397.3	85.9	180	11
SiO ₂ -Al ₂ O ₃ -MgO	×	60	272	99.7, 500	~85	~120	12
ZrO ₂ -SiO ₂	×	100	~200	97.5, 500	~75-85	~120	13
Al ₂ O ₃ /α-quartz	×	100	190	97.9, 500	~55	120	14
Mn/BaTiO ₃	×	20	~91	98.4, 2500	91 75.7	180 900	This work

* Only single-run anti-fouling test results are considered, excluding scenarios where membranes are cleaned and retested after fouling.

Comment 3: In Figure 2, how does the frequency of the alternating current affect the anti-fouling effect, and why doesn't the anti-fouling efficiency increase proportionally with the increase in AC frequency?

Response: Like the previous question, this one is very representative. Piezoceramic vibrations have an optimal resonance frequency, meaning the vibrations of the piezoceramic membrane do not simply increase with higher frequencies of the applied AC current. Thank you again for this insightful question. The related content is explained in the text as follows:

Line 81-85: Actually, from an ultrasonic perspective, the core component of an ultrasonic generator is the piezoelectric ceramic or piezoelectric crystal, and the built-in ultrasonic resonant frequency of piezoceramic membranes usually falls in the high-frequency ultrasonic range (>100 kHz). This high-frequency can generate significant ROS for advanced oxidation processes in water treatment, which previous studies overlooked but is potentially crucial for fouling control in piezoceramic membranes^{3, 14,}

Line 124-131: It was observed that within a certain range of the tested AC frequencies, membrane anti-fouling effectiveness did not proportionally increase with higher AC frequencies¹⁴. This suggests that the AC frequency of 265 kHz is optimal for generating substantial piezoelectric vibration in the Mn/BaTiO₃ membrane, leading to consistently high oil anti-fouling efficiency. Furthermore, increasing the intensity of AC, particularly at a resonance frequency of 265 kHz, amplified the vibrational amplitude of membrane. As a result, when the AC voltage was raised from 0 to 20 V, there was a corresponding enhancement in anti-fouling performance, ranging from 38.3% to 91.0% (Fig. 2b-c).

Line 225-228: Typically, piezoelectric materials exhibit optimal vibration performance at their specific resonant frequencies^{44, 45}. In this case, the Mn/BaTiO₃ membrane, known for its high piezoelectric properties, demonstrated its optimal resonant frequency around 265 kHz within the tested range.

Comment 4: Is there a possibility of co-sintering the BaTiO₃ piezoelectric ceramic membrane with other inorganic catalysts during the sintering process? This requires further explanation.

Response: Thank you for the reminder. Like most catalysts, piezoceramics are metal oxides, which allows them to be co-sintered together into piezoelectric catalytic membranes. Moreover, based on our discovery that piezoelectric vibrations can generate reactive oxygen species (ROS), your suggestion enables this work to extend from piezoelectric anti-fouling to the development of piezoelectric catalytic membranes. The related content has been supplemented in the text as follows:

Line 23-26: This offers profound insights into the distinct anti-fouling processes for organic and inorganic wastewater, supplementing and unifying the traditional singular vibrational anti-fouling mechanism of piezoceramic membranes, and potentially propelling the development of piezoelectric catalytic membranes.

Line 304-307: Additionally, by co-sintering functional catalysts with piezoelectric ceramics to form piezoelectric catalytic membranes, this approach not only combats membrane fouling but also advances catalytic membrane technology for wastewater treatment, with a particular focus on the treatment of persistent organic wastewaters.

Comment 5: At line 258, the generation of ROS is accompanied by the production of different free radicals; is there a possibility that H₂O₂ is produced in this process? The authors should conduct a deeper investigation into this mechanism.

Response: Thank you again for your constructive suggestion. Our subsequent experiments indeed verified the production of H₂O₂, and demonstrated that its generation is highly related to the vibration frequency of the membrane and the voltage of the applied alternating current. Additionally, based on the types of free radicals detected, we have proposed hypotheses regarding their intrinsic relationships with hydrogen peroxide. This new content has been added to the supplementary information, as follows:

Line 249-253 : As shown in Fig. 5b-c and Supplementary Fig. 11, upon stimulation at the optimal resonant high frequency (265 kHz, 20 V), significant ROS signals of ·OH, ¹O₂ and H₂O₂ were detected via the membrane built-in ultrasonic vibration. These ROS have redox potentials of 1.9~2.7 V for ·OH and 2.2 V for ¹O₂ versus the standard hydrogen electrode, sufficient to oxidize the organic foulants used in this study, thereby confirming the aforementioned hypothesis⁴⁸.

Supplementary Information:

Supplementary Fig. 11. H₂O₂ generated by the ultrasonic vibrations of the Mn/BaTiO₃ piezoelectric membranes under different AC conditions.

As illustrated in Supplementary Fig. 11, the membrane built-in ultrasonic vibration in water

generated H₂O₂, which is correlated with the applied AC frequency and voltage. The maximum production of H₂O₂ was achieved within 5 min at the optimal membrane resonance frequency of 265 kHz, reaching 65.6±5.7 μM L⁻¹. This result is consistent with the membrane anti-fouling performance. Based on the different ROS detected during the piezoelectric ultrasonic vibration membrane process, ROS interrelations could be proposed as:

REVIEWERS' COMMENTS

Reviewer #2 (Remarks to the Author):

The manuscript has been revised according to the reviewers' comments. I would recommend accept this manuscript

Reviewer #3 (Remarks to the Author):

The authors have well addressed my proposed concerns, and it could be published in the current form.